# Proregenerative Activity of IL-33 in Gastric Tissue Cells Undergoing *Helicobacter Pylori*-Induced Apoptosis

**DOI:** 10.3390/ijms21051801

**Published:** 2020-03-05

**Authors:** Weronika Gonciarz, Agnieszka Krupa, Magdalena Chmiela

**Affiliations:** Department of Immunology and Infectious Biology, Institute of Microbiology, Biotechnology and Immunology, Faculty of Biology and Environmental Protection, University of Lodz, 90-237 Lodz, Poland; weronika.gonciarz@biol.uni.lodz.pl (W.G.); agnieszka.krupa@biol.uni.lodz.pl (A.K.)

**Keywords:** *Helicobacter pylori*, gastric barrier, IL-33, proliferation, apoptosis

## Abstract

Interleukin (IL)-33 is a proinflammatory mediator that alerts the host immune system to disorders in tissue homeostasis. Aim. To understand the role of IL-33 in modulating gastric tissue cell growth affected by *Helicobacter pylori (H. pylori*). Methods. IL-33 production in guinea pigs *(Caviae porcellus*) experimentally infected with *H. pylori* was evaluated by ELISA or immunohistochemical staining. The proregenerative activity of IL-33 was evaluated using gastric epithelial cells and fibroblasts that were naive or transfected with IL-33 siRNA exposed to *H. pylori* glycine acid extract antigenic complex (GE), as well as by measuring cell migration, proliferation, metabolic activity and apoptosis. Animals infected by *H. pylori* responded with increased production of IL-33. Also, cells treated in vitro with GE released more IL-33 than cells that were unstimulated. Silencing IL-33 in cells resulted in downregulation of metabolic activity, adhesion, migration and proliferation, especially after treatment with *H. pylori* GE, as well as upregulation of cells apoptosis associated with caspase 3 increase and Bcl-xL decrease, suggesting proregenerative activity of IL-33. Interestingly, upregulation of cell proliferation by IL-33 was Erk independent. Our results indicate that IL-33 may protect gastric tissue from loss of homeostasis caused by deleterious effects of *H. pylori* components and the inflammatory response developed during infection.

## 1. Introduction

The gastric mucosa is exposed to endogenous factors such as hydrochloric acid, pepsin, and bile acids, as well as exogenous factors, including ethanol, nonsteroidal anti-inflammatory drugs and various microbial components [1,2]. Under physiological conditions, the mucous membrane maintains structural integrity due to pre-epithelial (phospholipid mucus phosphate), epithelial (carbohydrates, mucus, phospholipids, peptides, prostaglandins and heat shock proteins—Hsp) and postepithelial (hormones, growth factors and cytokines) defense mechanisms [3,4]. Microorganisms such as Gram-negative *Helicobacter pylori* (*H. pylori)* bacteria must overcome the mucous barrier to be able to reach epithelial cells and colonize the gastric epithelium (50% of the human population) [5,6]. The majority of *H. pylori*-infected individuals remain asymptomatic, but even in these subjects, infection increases the risk of serious diseases. A total of 10% to 15% of *H. pylori*-infected individuals develop chronic inflammatory responses, which correlate with an increased risk of gastric and duodenal ulcers, stomach cancer and mucosa-associated lymphoid tissue (MALT) lymphoma [7,8,9,10,11,12,13].Chronic colonization of the gastric mucosa by *H. pylori* and long-lasting interaction between bacteria and their soluble or cell-bound components as well as gastric epithelial cells determine the pathogenic process [10,14,15]. *H. pylori* urease neutralizes gastric juice acidity and affects the integrity of epithelial cell tight junctions [16]. Other virulence factors such as adhesins, gamma-glutamyl transpeptidase (GGT), neutrophil-activating factor (HP-NAP), vacuolating cytotoxin A (VacA), cytotoxin-associated gene A (Cag A) antigen and high temperature requirement A protein (HtrA) are involved in disease development [15,17,18,19,20,21]. *H. pylori* lipopolysaccharide (LPS) shows lower endotoxicity compared to that of the LPS of classic enteropathogens due to having a different lipid A structure [22,23,24,25]. However, due to the presence of Lewis (Le) determinants that mimic the host Le components and the ability to modulate the activity of immunocompetent cells, *H. pylori* LPS can help these bacteria evade the immune mechanisms of the host or induce autoimmune responses [26,27,28,29,30,31,32].

Tissue injury and epithelial barrier disfunction induced by *H. pylori* result in the secretion of danger signals by the host cells. Emergency molecules such as double-stranded DNA (dsDNA), high motility group box-1 protein (HMGB-1), ATP, cholesterol crystals, interleukin (IL)-1β, IL-33, heat shock proteins (Hsps), mitochondrial DNA or mitochondrial *N*-formyl peptides, which are called DAMPs (damage-associated molecular patterns), alert the innate immune system to changes in homeostasis [33,34]. IL-33 is a newly described member of the IL-1 family of cytokines that is a DAMP and it is produced by cells under physiological conditions and potentially during several diseases. A summary of major IL-33 activities is presented in Table 1 [35,36,37,38,39,40,41,42,43,44,45,46,47,48,49,50,51].

Increased expression of IL-33 has been shown in the gastric mucosa of patients infected with *H. pylori.* The level of IL-33 was higher in the acute phase of infection compared to that of the chronic phase [51]. Understanding the role of IL-33 in the pathogenesis of *H. pylori* infection requires further investigation. In our study, we used an experimental model of *H. pylori* infection in *Caviae porcellus* (guinea pigs) to examine by ELISA whether in vivo IL-33 was upregulated locally in gastric tissue homogenates and systemically in response to *H. pylori* infection. We also used cellular models of primary guinea pig gastric epithelial cells and fibroblasts, which are involved in wound healing in a subepithelial mucosa, for in vitro experiments [52,53]. By using these cells and by performing IL-33 silencing with siRNA, we examined how this cytokine influenced cell migration and proliferation, which are considered to be early biomarkers of cell regeneration activity. Cell migration and proliferation were evaluated in conjunction with cellular metabolic activity, shown as the ability of the cells to reduce 3-(4,5-dimethylthiazol-2-yl) and 2,5-diphenyltetrazolium bromide (MTT). Cell apoptosis was determined by terminal deoxynucleotidyl transferase (TdT)-mediated dUTP nick end labeling (TUNEL) assay, DNA damage was assessed by DAPI (4′,6-diamino-2-phenylindole) staining, and production of proapoptotic caspase-3 and antiapoptotic B cell lymphoma-extra-large (Bcl-xL) proteins was detected in cells by immunostaining. Activation/phosphorylation of extracellular signal-regulated kinase (pErk) was assessed in cells by using immunofluorescence.

## 2. Results

### 2.1. Production of IL-33 in Caviae porcellus colonized with H. pylori

The gastric tissue of guinea pigs inoculated with *H. pylori* was effectively colonized with the bacteria 7 and 28 days after the last inoculation, as shown by the staining of gastric tissue specimens with anti-*H. pylori* antibody conjugated to FITC (Figure 1A).

In the previous study performed on the same set of animals, we showed that gastric tissue colonization by *H. pylori* correlated with extended expression of mucin 5 (MUC5AC) and development of inflammation, which was acute after 7 days, whereas chronic inflammation occurred up to 28 days after the last inoculation [54]. Furthermore, infection was followed by upregulation of oxidative stress and cell apoptosis in the gastric tissue of *H. pylori*-infected animals, and this process was more pronounced at 7 days post inoculation compared to that at 28 days post inoculation, whereas cell proliferation was increased during the chronic phase of infection (28 days after inoculation), suggesting the initiation of gastric tissue regeneration processes [55].

In vivo experiments performed in the present study showed that *H. pylori* infection resulted in elevated production of IL-33. The level of IL-33 was increased locally in the gastric cells and gastric tissue homogenates of *H. pylori-*infected vs. uninfected animals at 7 and 28 days after the last inoculation (Figure 1B(i)–B(iii)), and systemically in the serum samples at 28 days post infection (Figure 1B(iv)).

### 2.2. IL-33 Production in Cell Cultures Exposed to H. pylori Components

After considering the potential role of IL-33 in modulating gastric tissue regeneration, we used primary guinea pig gastric epithelial cells and fibroblasts and performed transfections with siRNA to silence IL-33. The transfection efficiency was 92%, and the representative cells that were positive for FITC-conjugated siRNA are shown on Figure 2A.

Cells transfected with siRNA were examined for spontaneous and *H. pylori*-induced cell migration and proliferation in vitro conjunction with the ability to produce IL-33. Primary gastric epithelial cells and fibroblasts cultured in medium only produced the base level of IL-33, whereas cells treated with *H. pylori* GE produced significantly increased amounts of this cytokine, as detected intracellularly and in cell culture supernatants (Figure 2). Importantly, cells transfected with siRNA for IL-33 were not able to produce IL-33, regardless of their exposure to *H. pylori* GE (Figure 2).

### 2.3. Pro-Regenerative Activity of IL-33 in the Cell Cultures Exposed to H. pylori Components

Injury to the gastric mucosa in individuals infected with *H. pylori* triggers tissue regeneration processes. It was found that interaction between *H. pylori*-derived RpL1aa 2–20 peptide (Hp 2–20) and cellular formyl peptide receptors (FPRs) induces cell migration and proliferation, as well as the expression of vascular endothelial growth factor (VEGF), which is involved in tissue regeneration [56]. Millar et al., showed that the binding of IL-33 with its secretory receptor (ST2) is necessary for initiation of tissue healing [38]. It has also been suggested that IL-33 is involved in tissue regeneration by activating group 2 innate lymphoid cell (ILC2) to deliver amphiregulin, a protein that is essential for epidermal growth, cell survival and proliferation [57].

In our cellular models, the pro-regenerative activity of cells was evaluated based on cell migration, proliferation and the ability to reduce MTT. As shown in Figure 3A(i), control cells that were not transfected with IL-33 siRNA were able to migrate and minimize any scratches.

However, the wound healing ability of both types of cells was completely abrogated by siRNA silencing of IL-33, after up to 48 h of cell culture for gastric epithelial cells and up to 24 h of culture for fibroblasts, indicating that IL-33 potentiates cell migration, which is an early step in cell renewal (Figure 3).

Further, we examined the influence of *H. pylori* GE on the migration of cells with and without IL-33 siRNA transfection. Untransfected primary gastric epithelial cells that were treated with *H. pylori* GE migrated as effectively as those cells cultured in medium only, whereas IL-33 siRNA-transfected cells did not migrate in response to GE (Figure 3 A(i)). In comparison, the migration of untransfected fibroblasts was upregulated by GE compared to the movement of cells cultured in medium only. Importantly, cells transfected with IL-33 siRNA were unable to migrate at all (Figure 3A(ii)). These results indicate that the pro-regenerative potential of cells exposed to GE depends on the ability of cells to produce IL-33.

Cell proliferation plays a key role in the later stages of cell renewal. To determine the role of IL-33 driving cell proliferation in the presence of GE, we evaluated the proliferative activity of cells with and without IL-33 siRNA transfection that were unexposed or exposed to *H. pylori* GE. Only untransfected primary gastric epithelial cells and fibroblasts proliferated in response to GE (Figure 3B(i),B(ii)). The increased GE-induced cell proliferation was related to the increased effectiveness in MTT reduction that was observed in untransfected cells (Figure 3C(i),C(ii)), suggesting that the pro-regenerative activity of IL-33 is due to the upregulation of cell proliferation.

### 2.4. Control of H. pylori-Induced Erk Activation by IL-33

It has been shown that IL-33 enhances proliferation and invasiveness of decidual stromal cells by upregulating the chemokines CCL2/CCR2 through increasing the phosphorylation of nuclear factor NF-κB p65 and the extracellular signal-regulated kinase Erk [58]. In the present study, the level of phosphorylated Erk was increased in response to GE in primary gastric epithelial cells and fibroblasts that were untransfected or transfected with IL-33siRNA (Figure 4).

The level of phosphorylated Erk in IL-33 siRNA-silenced gastric epithelial cells treated with GE was significantly higher than that in untransfected cells (Figure 4A,B). This suggests that GE-induced Erk activation in gastric epithelial cells and fibroblasts is under the control of IL-33. The limitation of these data is associated with the lack of the IL-33 dependent molecular mechanism of Erk control, including the influence of well-defined *H. pylori* components and the inflammatory milieu. Further study in this area can help to better understand these processes.

### 2.5. Regulation of Cell Apoptosis by IL-33

We further explored whether the decreased migration, MTT reduction, and proliferation of cells transfected with IL-33 siRNA and exposed to *H. pylori* GE was a result of cell apoptosis. Seki et al., showed that IL-33 prevented cardiomyocytes from hypoxia-induced apoptosis both in vitro and in vivo and improved cardiac function and survival after myocardial infarction through ST2 (an IL-1 receptor family member) receptor signaling pathway [59]. However, IL-33 is sensitive to degradation and attenuation by effector caspases that are activated during apoptosis (caspase 3 and caspase 7) but is not a physiological substrate for the caspases associated with inflammation (caspase 1, 4 and 5) [60,61]. In our study, approximately 20% of untransfected gastric epithelial cells or fibroblasts cultured for 24 h in medium alone showed evidence of early apoptosis via TUNEL assay, whereas 40% of those cells underwent apoptosis in the GE treatment group (Figure 5).

Both cell types transfected with IL-33 siRNA had up to a 40% increase in apoptosis, indicating the role of IL-33 in controlling cell growth in medium alone. The percentage of IL-33 siRNA-transfected cells that were exposed to *H. pylori* GE that underwent apoptosis reached 60% and 70% in cell cultures of gastric epithelial cells and fibroblasts, respectively. These results show that GE alone promoted cell apoptosis and that IL-33 downregulated this process. In untransfected cells that were cultured in medium alone, the expression of the antiapoptotic protein Bcl-xL was higher than the expression of proapoptotic caspase 3; moreover, in cell cultures exposed to *H. pylori* GE, the ratio of Bcl-xL to caspase 3 was similar (Figure 6).

By comparison, in cultures of IL-33 siRNA-silenced cells incubated in medium only or in the presence of *H. pylori* GE, the level of proapoptotic caspase 3 was higher than the level of Bcl-xL (Figure 6A,B). The increased migration and proliferation of untransfected gastric epithelial cells and fibroblasts was possibly related to the ability of these cells to produce IL-33. Our data suggests that IL-33 prevents cell migration and proliferation by downregulating cell apoptosis.

## 3. Discussion

Colonization of the gastric mucosa by *H. pylori* in humans and in laboratory animals experimentally infected with *H. pylori* induces a chronic inflammatory response and injury to gastric tissue, which become massively infiltrated by immune cells to eliminate the infection and restore homeostasis. However, this excessive activation of immunocompetent cells may increase *H. pylori*-induced deleterious effects [62,63]. According to the “theory of danger”, necrotic cells release signaling molecules with adjuvant properties that activate innate immune cells and contribute to the development of specific adaptive immunity [47,64,65]. Emergency mediators are also released by apoptotic cells if these factors are not removed by phagocytes, which may occur during *H. pylori* infection, since *H. pylori* components downregulate phagocytotic processes and induce cell apoptosis in vivo and in vitro [66,67,68]. In the gastric mucosa of patients infected with *H. pylori*, during the acute phase of infection, enhanced expression of IL-33 suggests the role of this cytokine in activating immune cells and controlling gastric tissue homeostasis [51]. Further experiments are needed to specify the role of IL-33 during *H. pylori* infection.

In the present study, we asked whether *H. pylori-*induced gastric inflammation in guinea pigs that were experimentally infected with *H. pylori* correlated with the increased production of IL-33 and whether this cytokine was capable of promoting migration and proliferation of primary guinea pig gastric epithelial cells and fibroblasts in vitro.

The present study revealed that *H. pylori* in vivo and *H. pylori* components in vitro increased the production of IL-33. In response to *H. pylori*-induced gastric barrier injury, IL-33 may play a beneficial role in cell regeneration processes. However, excessive cell proliferation increases the risk of amplification of defective cells. In this case, the anti-apoptotic activity of IL-33 in gastric tissue exposed to carcinogenic *H. pylori* infection can be deleterious to the host due to promotion of neoplasia [6]. Krzysiek-Maczka et al. showed that *H. pylori* induces the differentiation of fibroblasts into myofibroblasts, increases expression of early carcinogenic markers such as hypoxia inducible factor (HIF)-1α and decreases expression of proapoptotic Bax [69]. Cell apoptosis may be considered as a mechanism for controlling the excessive proinflammatory activity of IL-33 bydelivering proapoptotic caspase-3, which can inactivate IL-33 [61]. Krzysiek-Maczka et al., have suggested that the over expression of heat shock protein Hsp-70 and the unchanged proliferation of fibroblasts in the gastric mucosa represent the enhanced protective activity of *H. pylori*-infected cells in maintaining their own integrity under inflammatory properties of these bacteria and bacterial-mediated cytotoxicity [69]. Previously, we showed that seven days after *H. pylori* inoculation of guinea pigs, the number of gastric tissue cells undergoing apoptosis was higher than that at 28 days after inoculation, which corresponded to acute and chronic infection, respectively. This was in line with an increased expression of antiapoptotic Bcl-xL in the gastric mucosa of *H. pylori*-infected guinea pigs after 28 days following inoculation [55]. In the present study, cells exposed to *H. pylori* GE responded by increasing apoptosis (Figure 5), which was correlated with an increased production of proapoptotic caspase 3 (Figure 6). In our previous study, which was carried out in an in vivo model of *H. pylori*-infected guinea pigs, increased production of metalloproteinase (MMP)-9 was detected, both locally and systemically [55]. Similarly, guinea pig gastric epithelial cells and fibroblasts exposed to *H. pylori* components in vitro responded by producing MMP-9 [55]. Our study showed that the level of phosphorylated Erk was increased in response to GE in primary gastric epithelial cells and fibroblasts (Figure 4A,B). Interestingly, the level of phosphorylated Erk in IL-33 siRNA-silenced gastric epithelial cells treated with GE was even higher than that in untransfected cells (Figure 4A,B), suggesting that GE-induced Erk activation in gastric epithelial cells and fibroblasts was under the control of IL-33. The increased activation of Erk in IL-33 siRNA-silenced cells observed in this study was potentially induced by MMP-9. Further study is needed to investigate whether IL-33 governs MMP-9 production.

In vivo, the concentration and activity of IL-33 can be modulated by the soluble IL-33 receptor ST2, which is delivered by fibroblasts and macrophages in response to bacterial LPS, as well as TNF-α, IL-1, or cytokines produced by Th2 lymphocytes. ST2 can form a complex with SIGIRR (Toll IL-1R8) and inhibit the biological activity of IL-33 [46]. It has been shown in *IL-33−/−* and *IL-33+/+* mice that IL-33 deficiency leads to delayed local inflammation by reducing neutrophil chemoattractant factors, resulting in delayed resolution of tissue damage during dextrin sodium sulfate-induced colitis [61]. The promotion of gastric epithelial cell apoptosis by *H. pylori* components, particularly LPS, may be a part of these destructive processes that increase gastric barrier permeability [67]. Yi et al., showed that LPS from *E. coli* induces a pathologic increase in lung vascular leakage under septic conditions due to increased apoptosis of microvascular endothelial cells [70]. Recently, increased permeability of in vitro monolayers of human gastric epithelial cells and endothelial cells in response to *H. pylori* LPS was observed in conjunction with elevated apoptosis [71].

Shibata et al., revealed that TNF-α, which is released by immunocompetent cells in patients with *H. pylori* infection, can promote cell apoptosis in the gastric mucosa. On the other hand, the amount of soluble TNF receptors (sTNF-Rs), which are expressed in the gastric mucosa, is significantly increased in *H. pylori-*positive patients, resulting in protection of gastric epithelial cells from TNF-α-induced apoptosis during *H. pylori* infection [72]. Due to this, the examination of sTNF-Rs as potential TNF-α modulators should be considered. In our guinea pig models, the concentration of systemic TNF-α in *H. pylori-*infected guinea pigs was lower than that in control animals.

Upregulation of programmed cell death during *H. pylori* infection can help to reduce microbial infection, separate infected cells from uninfected cells, eliminate mutated cells, and stimulate the immune system to attack the pathogen. However, upregulation of cell apoptosis may delay the process of tissue regeneration, particularly when the elimination of apoptotic cells is insufficient. During *H. pylori* infection, this could be due to *H. pylori*-induced diminished phagocyte activity [66,68]. Excessive cell apoptosis may also promote a chronic inflammatory response, which is potentially deleterious for the host.

## 4. Materials and Methods

### 4.1. Ethics Statement

In vivo experiments were developed according to the EU directive (Directive 2010/63/EU of the European Parliament and of the Council of 22 September 2010 on the protection of animals used for scientific purposes (Dz.U. L 276 z 20.10.2010, s. 33–79)), and were approved by the resolution of the Local Ethics Committee (LKE9) for Animal Experiments of the Medical University of Lodz, Poland, which was established by the Ministry of Science and Higher Education in Poland (Decision 58/ŁB45/2016).

### 4.2. H. pylori Infection in Caviae porcellus (Guinea Pigs)

Three-month-old, male Himalayan guinea pigs (400–600 g), free of pathogens, were housed in the Animal House at the Faculty of Biology and Environmental Protection, University of Lodz (Poland), kept in cages with free access to drinking water and fed with standard chow. The animals were inoculated per os with an *H. pylori* suspension as previously described [31,73]. Until day 7 or day 28 after inoculation, the health of the animals was monitored every day (body weight, water and food intake, behavioral symptoms, skin and fur condition, and diarrhea). Seven and 28 days after the last *H.* pylori inoculation, the animals were euthanized with an overdose of sodium barbiturate (Morbital, Biowet, Puławy, Poland), and the gastric tissue was collected for analyses, whereas blood samples were processed to obtain serum and then stored at −80 °C. *H. pylori* infection was confirmed by microscopic imaging of *Helicobacte*r-like organisms (HLOs), scoring tissue inflammation in thin layer preparations stained by routine histological staining, polymerase chain reaction (PCR) to detect *H. pylori cagA* and *ureC* gene sequences, and the laboratory enzyme-linked immunosorbent assay (ELISA) for anti-*H. pylori* IgG antibodies as previously described [55]. *H. pylori* bacteria were also visualized in gastric tissue specimens stained specifically with fluorescein isothiocyanate (FITC)-labeled anti-*H. pylori* antibody (MyBiosource, San Diego, CA, USA) diluted 1:200 and photographed with a confocal microscope (Leica TCS SP, Wetzlar, Germany), at magnifications of 10× and 40×. In total, 15 animals were used in the study: 5 control uninoculated animals and 10 animals infected with *H. pylori* (5–7 days infection and 5–28 days infection).

### 4.3. Cell Cultures

Primary guinea pig gastric epithelial cells were obtained as previously described [54]. Briefly, the stomach was isolated from the guinea pig, homogenized and treated with 0.25% trypsin (BIOMED-LUBLIN, Lublin, Poland) for 15 min at room temperature. The cell suspension at a density of 2 × 10^6^ cells/mL was seeded onto a 6-well plate (Becton Dickinson, Franklin Lakes, NY, USA) and cultured for 24 h (5% CO_2_, 37 °C) to allow the cells to adhere. The cells were cultured in a mixture of DMEM and Ham’s F-12 media (ratio 1:1; Sigma-Aldrich, Saint Louis, Mi, USA), supplemented with 10% fetal calf serum (FCS), 1% (*N*-2-hydroxyethylpiperazine-*N*-2-ethane sulfonicacid) (HEPES), penicillin (100 U/mL), streptomycin (100 µg/mL), amphotericin B (0.025 µg/mL), l-glutamine (2 mM/mL)epidermal growth factor (Sigma-Aldrich, Saint Louis, MI, USA) 0.01 µg/mL and 0.005% dexamethasone. The guinea pig fibroblasts (CRL-1405) were obtained from the American Type Culture Collection (ATCC, Rockville, Manassas, VA, USA) The cells were routinely grown as a monolayer in complete RPMI-1640 medium cRPMI(Sigma St. Louis, MI, United States) at 37 °C in a humidified atmosphere containing 5% CO_2_.Every 2–3 days, the medium was changed and the cells were passaged at 80–90% confluence.

### 4.4. Bacterial Stimuli

Glycine acid extract (GE) containing surface antigens from the reference *H. pylori* strain CCUG 17874 was obtained by extraction with 0.2 mol/L glycine buffer, pH 2.2, as previously described [74]. The protein composition was evaluated by SDS-PAGE and Western blotting (Milenia^®^ Blot *H. pylori*, DPC Biermann, GmbH, Bad Nauheim, Germany) with reference serum samples from patients infected with *H. pylori* [75]. The major proteins in GE that were recognized by sera from *H. pylori*-infected patients were: 120 kDa (CagA), 87 kDa (VacA), 66 kDa (UreB), 60 kDa (Hsp), 29 kDa (UreA), and 22–26kDa. The GE protein concentration was 600 μg/mL (NanoDrop 2000c Spectrophotometer, Thermo Scientific, Waltman, WY, United States), which was adjusted to 10 µg/mL for experiments. The GE contained <0.001 EU/mL of LPS, as shown by the chromogenic Limulus amebocyte lysate test (Lonza, Braine-Alleud, Belgium). The antigen concentrations were adjusted experimentally or adopted from previously performed experiments [27,28,31,32,67].

### 4.5. IL-33 siRNA Silencing

Primary guinea pig gastric epithelial cells and fibroblasts were seeded on 6-well plates containing glass coverslips at the density of 5 × 10^5^ cells/well in 1 mL of RPMI-1640 or F12:DMEM (Sigma-Aldrich, Saint Louis, MI, USA), both antibiotic free, supplemented with 10% FCS and cultured until complete confluence (37 °C, 5%CO_2_). Next, the cells were transfected with theIL-33 siRNA (Santa Cruz Biotechnology, Santa Cruz, CA, USA) according the manufacturer’s protocol. Briefly, the cells were washed with transfection medium and then incubated with the transfection mixture containing IL-33siRNA, siRNA transfection reagent, and siRNA transfection medium for 6 h (37 °C, 5%CO_2_). After 6 h, the cells were supplemented with culture medium and incubated until they were used (within 24 h). Then, the cells were stimulated for 24 h with *H. pylori* GE or medium only. Control FITC-conjugated siRNA was used to calculate the efficiency of transfection (percentage of positively transfected cells). A minimum of 100 cells was analyzed. Moreover, the effectiveness of silencing (IL-33 production) was confirmed in cell culture supernatants by ELISA (MyBiosource, San Diego, CA, USA) or intracellularly using microscopy and rabbit polyclonal primary anti-IL-33 antibodies (Thermo Fisher Scientific, Waltam, MA, USA) followed by secondary FITC-conjugated goat anti-rabbit antibody (Boster Bio, Pleasanton, CA, USA). Cell nuclei were visualized with 4′,6-diamidino-2-phenylindole (DAPI) (Sigma-Aldrich, Saint Louis, MI, USA),). Stained samples were viewed using a fluorescence microscope (Zeiss, Axio Scope, A1, Jena, Germany) at appropriate wavelengths (FITC (excitation 495 nm, emission 519 nm) and DAPI (358 nm excitation, 461 nm emission) at 200× magnification, and the fluorescence intensity was measured using ImageJ software version 1.48v (National Institute of Health, Bethesda, MD, USA). Four independent experiments were carried out with three replicates for each experimental variable.

### 4.6. Evaluation of Cell Migration in a Scratch Wound Healing Assay

Cell migration was evaluated in vitro by a scratch wound assay using primary gastric epithelial cells or fibroblasts with and without IL-33 siRNA silencing that were untreated or treated with *H. pylori* GE antigenic complex, as previously described [67]. The cells were seeded on 6-well plates at a density of 5×10^5^ cells/well in 1 mL of cRPMI-1640 or cF12:DMEM medium, supplemented with 2% FCS and 1% penicillin and streptomycin (Sigma-Aldrich, Saint Louis, MI, USA), and cultured until complete confluence. Next, the cell monolayers were scratched with a sterile 200 µL pipette tip (designated as time 0 of wound repair), and then *H. pylori* GE was added to the cells. Twenty-four hours later, the *H. pylori* components were removed, and the cells were washed twice with culture medium. Wound images were taken at 0, 24, and 48 h by a digital camera (Nikon P20, Tokyo, Japan). The scratched areas were measured using ImageJ software version 1.48v (National Institute of Health, Bethesda, MD, USA). Each wound was measured four times. Wound healing in the presence of the bacterial components is expressed as a percentage of cells migrating to the wound zone in comparison to that of the untreated cells. Four independent experiments were carried out with three replicates for each experimental variable.

### 4.7. Cell Viability Assays

Metabolic activity of untransfected or IL-33 siRNA-transfected cells that were untreated or treated with *H. pylori* components was evaluated colorimetrically using thetetrazolium yellow dye MTT (3-(4,5-dimethylthiazol-2-yl)-2,5-diphenyltetrazolium bromide), which is reduced by living cells to yield soluble purple formazan crystals, as previously described [67]. Absorbance at 570 nm was estimated with a Victor 2 plate reader (Wallac, Oy, Turku, Finland). The results of at least four independent experiments performed in triplicate for each experimental variable are presented as the median percentage with the range, relative to untreated cells. The effectiveness of MTT reduction was calculated based on the following formula:MTT reduction relative to untreated cells (%) = (absorbance of treated cells/absorbance of untreated cells × 100%) − 100%

The ability of cells to proliferate was examined by a radioactive proliferation assay based on the measurement of tritiated thymidine (3H-TdR) incorporated during DNA synthesis as previously described [67]. The incorporation of thymidine was measured using a MicroBeta 2 scintillation counter (Wallac Oy, Turku, Finland) after harvesting the cells on fiber filters. The results are expressed as median counts per minute (cpm)/culture with the range. Four independent experiments were performed in triplicate. The stimulation index (SI), expressed as the relative cpm ratio, was calculated by dividing the counts/min for the cell cultures with stimulation by the cpm counts/min for the cell cultures without stimulation. SI values higher than 1.0 (cut-off) were considered to be a positive result in the proliferation assay.

### 4.8. Cell Apoptosis

Apoptosis of untransfected or IL-33 siRNA-transfected primary guinea pig gastric epithelial cells and fibroblasts was evaluated by TUNEL assay (Cell Meter™ TUNEL apoptosis assay kit, AAT Bioques, Sunnyvale, USA) according to the manufacturer’s instructions. Briefly, after 24 h of stimulation with *H. pylori* GE or culture in medium only, the cells were fixed with 4% formaldehyde, permeabilized with 0.02% TritonX-100 in PBS and stained with Tunnelyte^TM^ red dye diluted 1:100 for 1 h at 37 °C and 5% CO_2_. The cell nuclei were counterstained with Hoechst stain (Cell Meter™ TUNEL apoptosis assay kit, AAT Bioques, Sunnyvale, USA) diluted 1:1000 in PBS for 15 min at room temperature. Cells with apoptotic changes were imaged using a fluorescence microscope (Zeiss, AxioScope, A1, Jena, Germany) at a wavelength of 550 nm (excitation) and 590 nm (emission) and a magnification of 20/40×. Four independent experiments were carried out with three replicates for each experimental variable. Furthermore, cells that were fixed with 4% formaldehyde and permeabilized with 0.2% Triton X-100 in PBS were stained overnight at 4 °C with primary anti-Bcl-xL antibodies or with primary anti-cleaved Caspase 3 antibodies (Cell Signaling Technology, Danvers, MA, USA), diluted 1:500 in 1% BSA/PBS, followed by incubation with FITC-conjugated secondary antibodies (Invitrogen Carlsbad, CA, USA) diluted 1:2000 in 1% BSA/PBS. Cell nuclei were counterstained with DAPI (Sigma-Aldrich, Saint Louis, MI, USA), ) diluted 1:1000 in PBS for 15 min at room temperature. Stained cells were observed in fluorescent microscope at appropriate wavelengths: FITC (excitation 495 nm, emission 519 nm) and DAPI (excitation358 nm, emission461 nm) at 1000× magnification. Fluorescence intensity was measured using ImageJ software version 1.48v via the fluorescence microscope [76] (Zeiss, Axio Scope, A1, Jena, Germany) Four independent experiments were carried out in three replicates for each experimental variable.

### 4.9. Erk Activation/Phosphorylation (pErk)

Primary gastric epithelial cells and fibroblasts were placed on 96-well culture plates (Thermo Fisher Scientific, Waltam, MA, USA) at a density of 1 × 10^5^ cells/well (volume 100 µl) in DMEM:F12 or RPMI-1640 medium, respectively (37 °C, 5% CO_2_). Unstimulated (control) or *H. pylori* GE-stimulated cells (concentration 10 μg/mL) were incubated for 1, 3, 6, 24, and 48 h, fixed with 4% formaldehyde solution for 20 min, and then washed 3 times in PBS, followed by incubation with 0.02% TritonX-100 for 10 min to permeabilize the membranes. Next, the cells were incubated with 3% BSA in PBS to block nonspecific binding and with primary rabbit anti-phospho-Erk 1/2 (Thr202/Tyr204)antibody(Cell Signaling Technology, Danvers, MA, USA) ) at a 1:200 dilution, followed by incubation with FITC-conjugated chicken anti-rabbit secondary antibody (Thermo Fisher Scientific, Waltam, MA, USA). Three independent experiments were performed in triplicate. Erk activation was assessed quantitatively based on the green fluorescence intensity that was measured using a multifunctional Victor 2 reader (Wallac, Oy, Turku, Finland) at the following wavelengths for FITC: 495 nm (excitation) and 519 nm (emission).

### 4.10. Statistical Analysis

All values are expressed as the median values with a range. The differences between tested variables were assessed using Statistica 12 PL software with a nonparametric Mann-Whitney U test or Kruskal-Wallis test. The results were considered statistically significant when *p* < 0.05.

## 5. Conclusions

In the light of the results obtained in this study, increased production of IL-33 in response to *H. pylori* and components of this bacterium could be considered as an important host mechanism promoting gastric tissue renewal due to IL-33-induced cell migration and proliferation and downregulation of *H. pylori*-induced cell apoptosis. This indicates that IL-33 may play a beneficial role in cell regeneration processes during *H. pylori* infection. However, excessive cell proliferation increases the risk of amplification of defective cells. In such cases, upregulation of anti-apoptotic activity of IL-33 in gastric tissue exposed to carcinogenic *H. pylori* potentially can promote gastric neoplasia (Figure 7).

Figure 7 visualizes the possible mechanisms of IL-33 in *H. pylori* induced pathogenesis. Thus, colonization of gastric mucosa with *H. pylori* results in tissue injury and loss of local homeostasis. These events upregulate secretion of IL-33, which alarms the host immune system. Activated immune cells deliver MMP-9, which promotes apoptosis of gastric epithelial cells. Enhancement of apoptosis may help to control inflammatory response and amplification of defective cells. Moreover, loss of gastric epithelial cells should be replenished to prevent the gastric barrier integrity. In this case, increased secretion of IL-33 in the gastric tissue colonized with *H. pylori* seems to be correlated with regeneration processes due to downregulation of apoptosis. However, such IL-33 activity may potentially promote the development of gastric cancer induced by *H. pylori* infection.

## Figures and Tables

**Figure 1 ijms-21-01801-f001:**
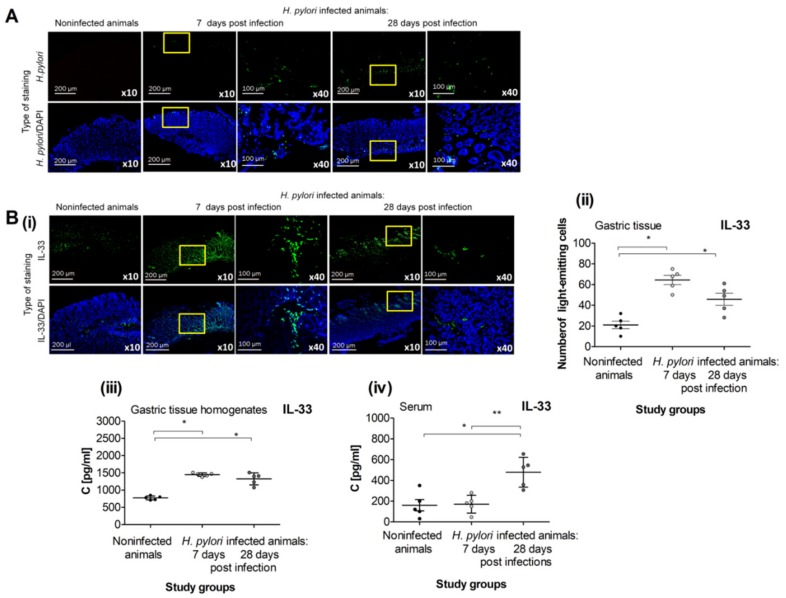
IL-33 in *Caviae porcellus* colonized with *H. pylori*. (**A**) Representative images of gastric tissue of guinea pigs noninfected or infected with *H. pylori,* 7 and 28 days from inoculation, stained with anti-*H. pylori* antibodies conjugated with FITC (fluorescein isothiocyanate; green) and counterstained with DAPI (4′,6-diamino-2-phenylindole; blue), photographed in the confocal microscope (Leica TCS SP, Wetzlar, Germany), at magnification ×10 or ×40. (**B**) Production of IL-33 in *H. pylori* infected vs. noninfected guinea pigs. (**i**) Representative images of gastric tissue of noninfected or *H. pylori* infected guinea pigs, 7 and 28 days from inoculation, stained with FITC-conjugated anti-IL-33 antibody (green) and counterstained with DAPI (blue), photographed in the confocal microscope (LeicaTCS SP, Wetzlar, Germany), at magnification ×10 or ×40. (**ii**) Level of intensity of IL-33 in the gastric tissue sections of noninfected or *H. pylori* infected guinea pigs, 7 and 28 days from inoculation, stained with FITC-conjugated anti-IL-33 antibody, assessed using the Image J software version 1.48v (National Institute of Health, United States). (**iii**) Level of IL-33 in gastric tissue homogenates of noninfected or *H. pylori* infected guinea pigs, 7 and 28 days from the last inoculation, estimated by using the ELISA assay. (**iv**) Level of IL-33 in serum samples of noninfected or *H. pylori* infected guinea pigs, 7 and 28 days from inoculation, estimated by using the ELISA assay. Results are showed as the median from the range of five experiments performed in triplicates for each experimental variant. Statistical analysis was performed using the nonparametric U Mann-Whitney test with significance *, ** *p* < 0.05 (* non-infected versus infected animals 7 or 28 days after inoculation; ** animals 7 days versus 28 days after inoculation).

**Figure 2 ijms-21-01801-f002:**
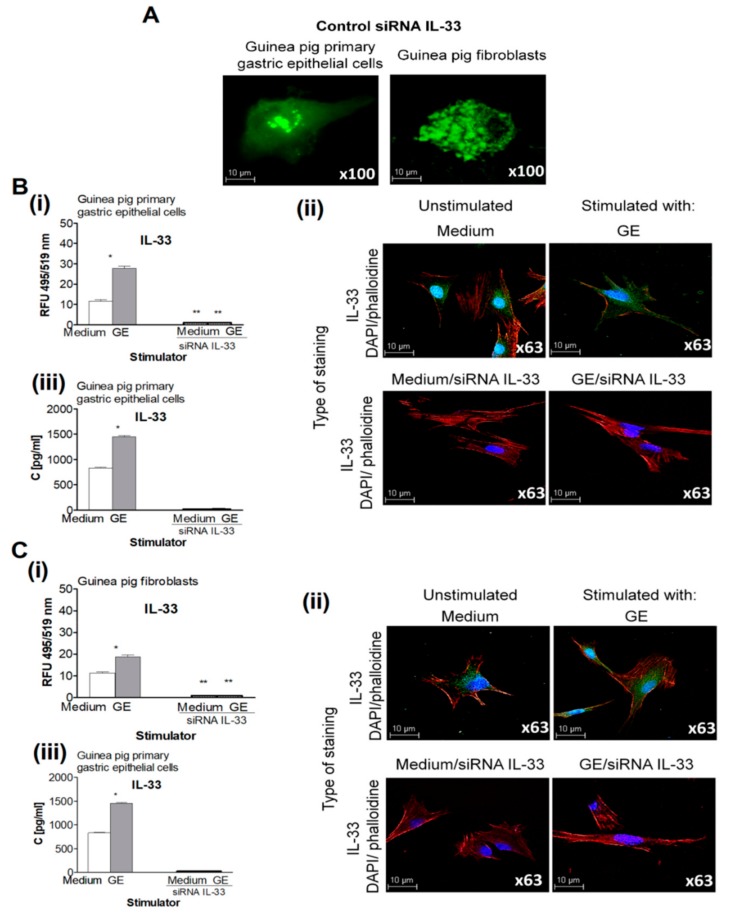
Production of IL-33 in cell cultures exposed to *H. pylori* components. (**A**) Representative images of guinea pig primary gastric epithelial cells or fibroblasts treated with control siRNA- labelled with fluorescein isothiocyanate; green (FITC) and photographed in the confocal microscope (Leica TCS SP Wetzlar, Germany), at magnification ×100. (**B**) Production of IL-33 by the guinea pig primary gastric epithelial cells not transfected or transfected with siRNA IL-33, unexposed or exposed on *H. pylori* glycine acid extract (GE). (**i**) Level of intracellular IL-33 in the guinea pig primary gastric epithelial cells measured as relative fluorescence units (RFU). (**ii**) Representative images of the guinea pig primary gastric epithelial cells stained for intracellular IL-33 (green) and counterstained with 4′,6-diamino-2-phenylindole (DAPI; blue) or phalloidin –Texas Red^TM^ (red), photographed in the confocal microscope (Leica TCS SP, Wetzlar, Germany), at magnification ×63. (**iii**) Concentration of IL-33 in the supernatants of guinea pig primary gastric epithelial cells estimated by the enzyme-linked immunosorbent assay (ELISA). (**C**) Production of IL-33 by the guinea pig fibroblasts, not transfected or transfected with siRNA IL-33, unexposed or exposed on *H. pylori* GE. (**i**) Level of intracellular IL-33 in the guinea pig fibroblasts measured as relative fluorescence units (RFU). (**ii**) Representative images of the guinea pig fibroblasts stained for intracellular IL-33 (green) and counterstained with DAPI (blue) or phalloidin (red) and photographed in the confocal microscope (Leica TCS SP, Wetzlar, Germany) at x63 magnification. (**iii**) Concentration of IL-33 in supernatants of guinea pig fibroblasts estimated by the ELISA. Results are showed as median with the range of five experiments performed in triplicates for each experimental variant. Statistical analysis was performed using the nonparametric U Mann-Whitney test with significance for *, ** *p* < 0.05,* unstimulated vs. GE stimulated untransfected cells, ** untransfected vs. transfected cells in medium only or in medium with GE.

**Figure 3 ijms-21-01801-f003:**
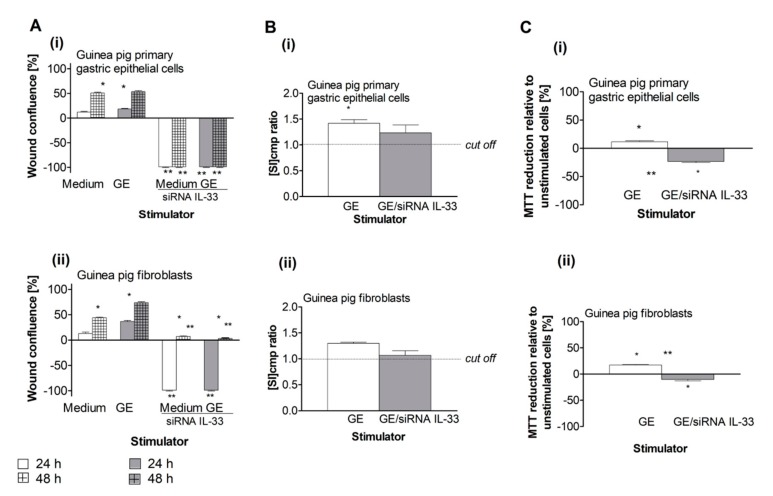
Pro-regenerative activity of IL-33 in the cell cultures exposed to *H. pylori* components. (**A**) Migration effectiveness of the guinea pig primary gastric epithelial cells (**i**) or fibroblasts (**ii**), not transfected or transfected with siRNA IL-33, cultured in medium only or in the milieu of *H. pylori* glycine acid extract (GE), expressed as the percentage of wound confluence in the scratch assay. (**B**) Proliferation of the guinea pig primary gastric epithelial cells (**i**) or fibroblasts (**ii**) untransfected or transfected with siRNA IL-33 in medium only or in the milieu of *H. pylori* GE, estimated in a radioactive assay. Proliferation is shown as stimulation index (SI), which was calculated, expressed as cpm ratio of unstimulated vs GE stimulated cells. (**C**) Metabolic activity of the guinea pig primary gastric epithelial cells (**i**) or fibroblasts (**ii**), not transfected or transfected with siRNA IL-33, expressed as the 3-(4,5-dimethylthiazol-2-yl)-2,5-diphenyltetrazolium bromide (MTT) reduction by stimulated cells relative to unstimulated cells. Results are showed as a median with a range of five experiments performed in triplicates for each experimental variant. Statistical analysis was performed using the nonparametric U Mann-Whitney test with significance for *, ** *p* < 0.05, *** unstimulated (only medium) vs. GE stimulated, untransfected or tranfected cells, ** untransfected vs. transfected cells in medium only or in medium with GE.

**Figure 4 ijms-21-01801-f004:**
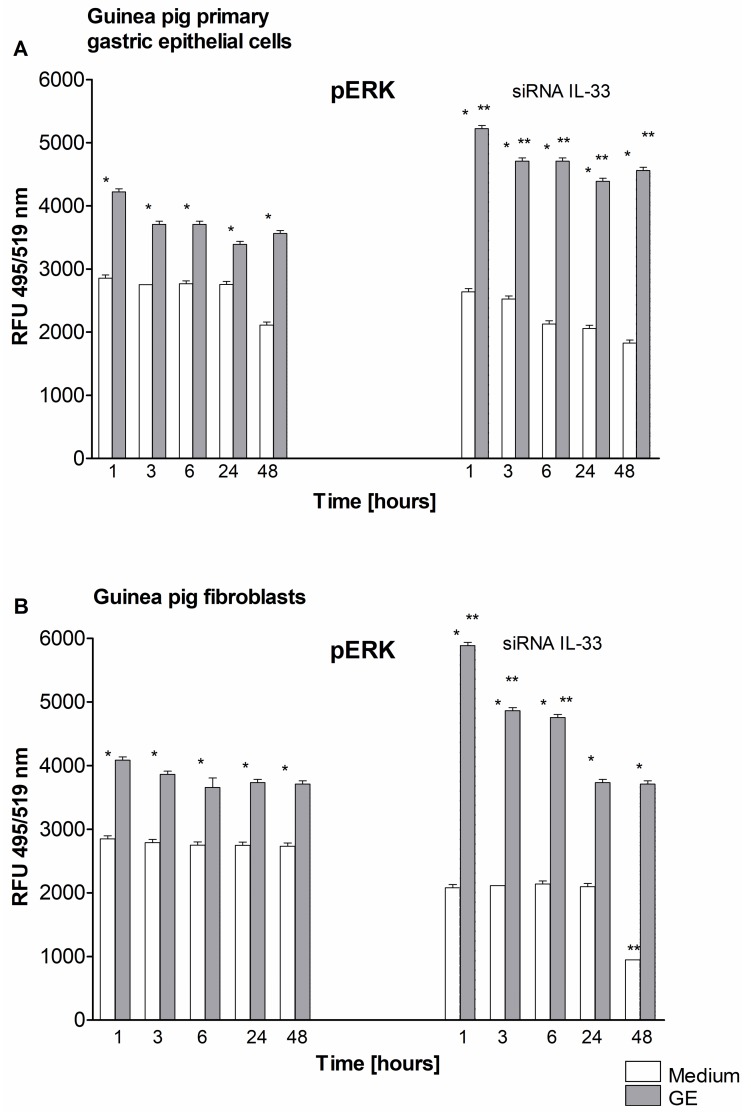
Activation/phosphorylation of Erk. Level of phosphorylated Erk (pERK) detected at time points: 1, 3, 6, 24, 48 h in cell cultures of guinea pig primary gastric epithelial cells (**A**) or fibroblasts (**B**), not transfected or transfected with siRNA IL-33, in medium only or exposed to *H. pylori* glycine acid extract. (GE). Activation of Erk was estimated based on relative fluorescence units (RFU) of cells intracellularly stained for pErk. Fluorescence was measured in a fluorescence reader (Victor 2, Wallac, Oy Turku, Finland) at wavelength: 495 excitation and 519 emission (emission). The results of four independent experiments performed in triplicates for each experimental variant are presented. Statistical analysis was performed using the nonparametric U Mann-Whitney test with significance for *, ** *p* < 0.05, *** unstimulated (only medium) vs. GE stimulated untransfected or transfected cells, ** untransfected vs. transfected cells in medium only or in medium with GE.

**Figure 5 ijms-21-01801-f005:**
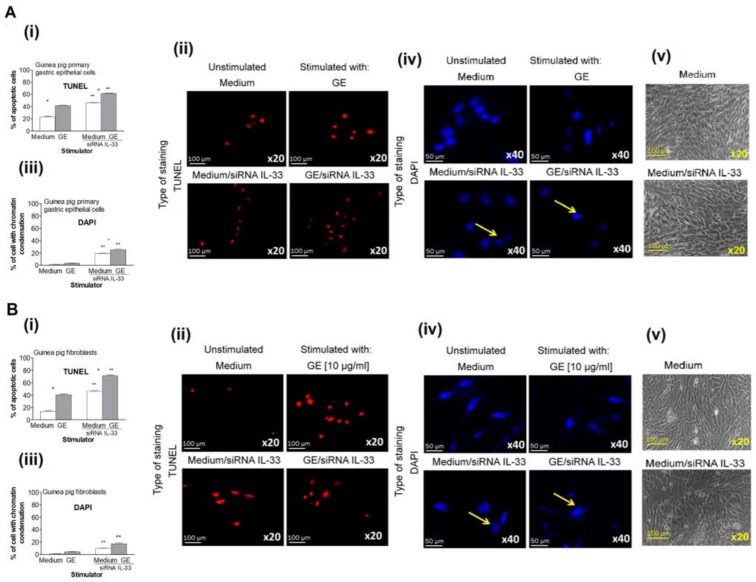
Assessment of apoptosis by terminal deoxynucleotidyl transferase (TdT)-mediated dUTP nick end labeling (TUNEL) assay and 4′,6-diamino-2-phenylindole (DAPI) nuclear staining. Results for the guinea pig primary gastric epithelial cells (**A**) or fibroblasts (**B**), untransfected or transfected with siRNA, in medium only or exposed to *H. pylori* glycin acid extract) (GE)presented as the percentage of apoptotic cells stained in the TUNEL assay and counted using the Image J software version 1.48v (National Institute of Health, United States) (**i**). (**ii**) Representative images of cells stained in the TUNEL assay (red nuclei) and photographed in the fluorescence microscope (Axio Scope A1, Zeiss, Germany) at wavelengths: 550 nm (excitation) and 590 nm (emission), at magnification x20. (**iii**) The percentage of cells stained with DAPI, indicating the chromatin condensation. (**iv**) Representative images of cells stained with DAPI (blue nuclei) and photographed in the fluorescence microscope (Axio Scope A1, Zeiss, Germany) at wavelengths: 345 (excitation) and 455 nm (emission), at magnification ×40. (**v**) Representative images of untransfected cells cultured in medium only (medium) or transfected with siRNA IL-33 and cultured in medium only (medium/siRNA IL-33) photographed in the phase-contrast microscope (YS100 Nikon, Tokyo, Japan), at magnification ×20. The results of four independent experiments performed in triplicates for each experimental variant are presented. Statistical analysis was performed using the nonparametric U Mann-Whitney test with significance for *, ** *p* < 0.05, * unstimulated (only medium) vs. GE stimulated, untransfected or tranfected cells, ** untransfected vs. transfected cells, in medium only or in medium with GE.

**Figure 6 ijms-21-01801-f006:**
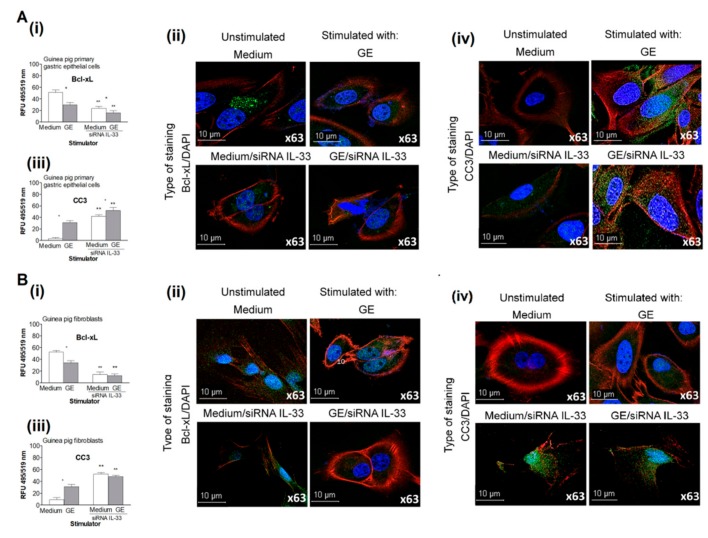
The expression of anti-apoptotic Bcl-xL and cleaved caspase 3 in cell cultures in vitro. Levels of anti-apoptotic Bcl-xL and cleaved caspase 3 (CC3) in cell cultures of guinea pig primary gastric epithelial cells (**A**) or fibroblasts (**B**), untransfected or transfected with siRNA cultured in medium only or exposed to *H. pylori* glycine acid extract (GE ).The level of intensity of Bcl-xL (**i**) and CC3 (**iii**) was measured using the Image J software version 1.48v (National Institute of health, United States). Representative images of cells stained for Bcl-xL (**ii**) or CC3 (**iv**) and photographed under 63x magnification in confocal microscope (LeicaTCS SP, Wetzlar, Germany)). 4′,6-diamino-2-phenylindole (DAPI) was used for nuclear staining and phalloidin –Texas Red^TM^ for cytoskeleton staining. The results of four independent experiments performed in triplicates for each experimental variant are presented. Statistical analysis was performed using the nonparametric U Mann-Whitney test with significance for *, ** *p* < 0.05, *** unstimulated (only medium) vs. GE stimulated, untransfected or tranfected cells, ** untransfected vs. transfected cells in medium only or in medium with GE.

**Figure 7 ijms-21-01801-f007:**
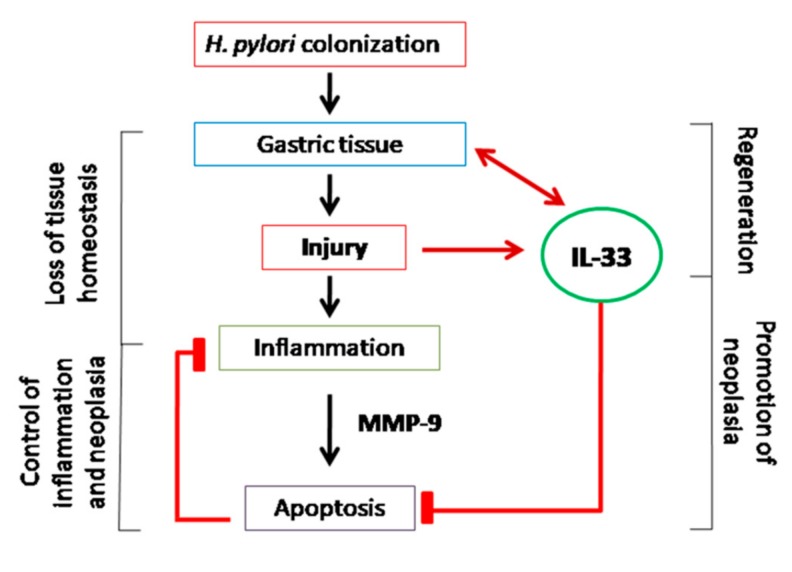
Possible mechanisms of IL-33 in *H. pylori-*induced pathogenesis.

**Table 1 ijms-21-01801-t001:** Summary of major IL-33 activities.

Function	Activity	Ref.
DAMP	signals the innate immune cells in response to stress or cell membrane disruption	[33,34]
Transcription factor	regulates the gene expression	[35,36,37,38]
Cytokine	signals the immune cells via ST2 receptor to eliminate infection and manage tissue injury (proregenerative activity)	[39,40,41,42]
	modulates the T helper 2 (Th2) lymphocytes as well as Th1 and regulatory lymphocytes	[43,44,45]
	upregulates the inflammatory response to bacterial LPS	[42]
	potentially is involved in the development of disease due to maintenance of chronic inflammation	[38,44,45,46,47,48,49,50,51]

DAMP (damage-associated molecular pattern); LPS (lipopolysaccharide); ST2 (secretory receptor, an interleukin (IL)-1 receptor family member); Th (T helper lymphocytes).

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
