# Peer review of "Proregenerative Activity of IL-33 in Gastric Tissue Cells Undergoing *Helicobacter Pylori*-Induced Apoptosis"

_ijms, 2020, doi:10.3390/ijms21051801_

Round 1

Reviewer 1 Report

The authors present and interesting topic here on Proregenerative Activity of IL-33 in Gastric Tissue Cells Undergoing Helicobacter Pylori-Induced Apoptosis. A well conducted research. Issues which will need to be addressed are:

Abstract look ok.

Introduction: would be better if you can show a diagram regarding the pathogenesis or mechanism or gastric mucosa and IL-33

Materials and methods section should be before results and discussion

In methods section please explain in confounders to the study and how they were resolved.

Discussion section:

Explain the limitations of the study

Conclusion section should be included

Figures and tables look ok

English and grammar needs to be thoroughly checked throughout the manuscript

Author Response

Reviewer 1

R1. “Introduction: would be better if you can show a diagram regarding the pathogenesis or mechanism or …. gastric mucosa and IL-33”

Responses: Table 1 summarizing major IL-33 activities and Fig 7 showing the possible mechanisms of IL-33 role in H. pylori induced pathogenesis were included in the Introduction and Conclusions sections, respectively.

Add in 94 line

R1. Materials and methods section should be before results and discussion.

Responses: Materials and methods section is right after Discussion and before Conclusions according to the journal requirements

Add in 16 line

R1: In methods section please explain in confounders to the study and how they were resolved

Response: Potential confounders to the study were associated with methodology and were resolved in preliminary study

R1: Discussion section: Explain the limitations of the study

Response: In the section Results and Discussion 2.4 Control of H. pylori-induced Erk activation by IL-33.

The limitation of these data is associated with missing the IL-33 dependent molecular mechanism of Erk control, including the influence of well-defined H. pylori components and the inflammatory milieu. Further study in this area can help to better understand these processes.

Add in 243 line

R1: Conclusion section should be included

Response: Conclusion section was included to the text

Add in 654 line

R1: English and grammar needs to be thoroughly checked throughout the manuscript

Response: The linguistic correction of the manuscript was performed, and the text was revised. We also include the language certificate made by American Journal Experts PDF documents

Reviewer 2 Report

The presented manuscript “Proregenerative Activity of IL-33 in Gastric Tissue Cells Undergoing Helicobacter Pylori-Induced Apoptosis” is an interesting topic and could be useful to understand the role of IL-33 in H pylori infection. The manuscript is well written and discussed. Though, I have a few minor comments.

  • Section 2 results and discussion. TNF-alpha data has not shown to support the sentence " In the present study, primary guinea pig gastric epithelial cells and fibroblasts that were untransfected or transfected with IL-33 siRNA did not produce TNF-α in response to stimulation with H. pylori GE”
  • The statistics in figure 3A ii, C i, ii is confusing. How the two different stars (single and double stars) represent two bars? please clarify it.
  • Figure 2A please replace “contriol” with control. Please also complete the text “Guinea pig primary gastric cells of”?
  • Figure 4A, please replace “epothelial” with “epithelial”.

Author Response

Reviewer 2.

R2: Section 2 results and discussion. TNF-alpha data has not shown to support the sentence “In the present study, primary guinea pig gastric epithelial cells and fibroblasts that were untransfected or transfected with IL-33 siRNA did not produce TNF-α in response to stimulation with H. pylori GE”

Response: This section of Results and Discussion was revised to support and explain the observation

R2: The statistics in figure 3A ii, C i, ii is confusing. How the two different starts (single and double stars) represent two bars? Please clarify it.

Response: The explanation of statistic in Figure 3 and following figures was included in figure legends

R2: Figure 2A please replace” contriol” with control. Please also complete the text “Guinea pig primary gastric cells of”?

Response: Misspelled word was corrected, and the part of text completed

R2: Figure 4A, please replace “epothelial with “epithelial”

Response: Misspelled word in the text was corrected

Language Certificate was add as a PDF file
